



# Air-sea interactions in stable atmospheric conditions: Lessons from the desert semi-enclosed Gulf of Eilat (Aqaba)

Shai Abir[1,2], Hamish A. McGowan[3], Yonatan Shaked[1, 4], Hezi Gildor[1], Morin [1]Efrat, Nadav G. Lensky[1,2]

[1] The Hebrew University of Jerusalem, Jerusalem, Israel.

[2] Geological Survey of Israel, Jerusalem, Israel.

[3] Atmospheric Observations Research Group, The University of Queensland, Brisbane, Australia.

[4] Interuniversity Institute for Marine Sciences, Eilat, Israel.

*Correspondence to*: Nadav G. Lensky (nadavl@gsi.gov.il)

**Plain Language Summary**

Understanding air-sea heat and gas exchange is vital for modeling ocean and atmosphere dynamics. Direct measurements of heat fluxes over the Gulf of Eilat revealed a 3.22 m year$^{-1}$ evaporation rate, almost twice than the widely used bulk formulae estimations; this discrepancy is attributed to the stability of the atmospheric boundary layer in desert marine environments. Surface heat fluxes cool on an annual scale the gulf water, balanced

by currents that carry heat from the Red Sea to the gulf.

**Abstract.** Accurately quantifying air-sea heat and gas exchange is crucial for comprehending thermoregulation processes and modeling ocean dynamics; these models incorporate bulk formulae for air-sea exchange derived in unstable atmospheric conditions. Therefore, their applicability in stable atmospheric conditions, such as desert-

enclosed basins in the Gulf of Eilat/Aqaba (coral refugium), Red Sea, and Persian Gulf, is unclear. We present 2-year Eddy Covariance results from the Gulf of Eilat, a natural laboratory for studying air-sea interactions in stable atmospheric conditions which are directly related to ocean dynamics.

The measured mean evaporation, 3.22 m year$^{-1}$, approximately double than previously estimated by bulk formulae, is exceeding the heat flux provided by radiation. Notably, in arid environments wind speed seasonal trend is

compelling maximum evaporation in summer, with minimum winter rate. The higher evaporation rate appears when elevated wind, particularly in the afternoon, coincide with an increase in vapor pressure difference. The bulk formulae approach inability to capture the seasonal (opposite from our measurements) and annual trend of evaporation is linked to errors in quantifying of the atmospheric boundary layer stability parameter.

Most of the year, there is a net cooling effect of surface water (-79 W m$^{-2}$), primarily through evaporation. The

substantial heat deficit is compensated by the advection of heat via northbound currents from the Red Sea, which we indirectly quantify from energy balance considerations. Cold and dry synoptic-scale winds induce extreme heat loss through air-sea fluxes, and are correlated with destabilizing of the water column during winter and initiating vertical water column mixing.






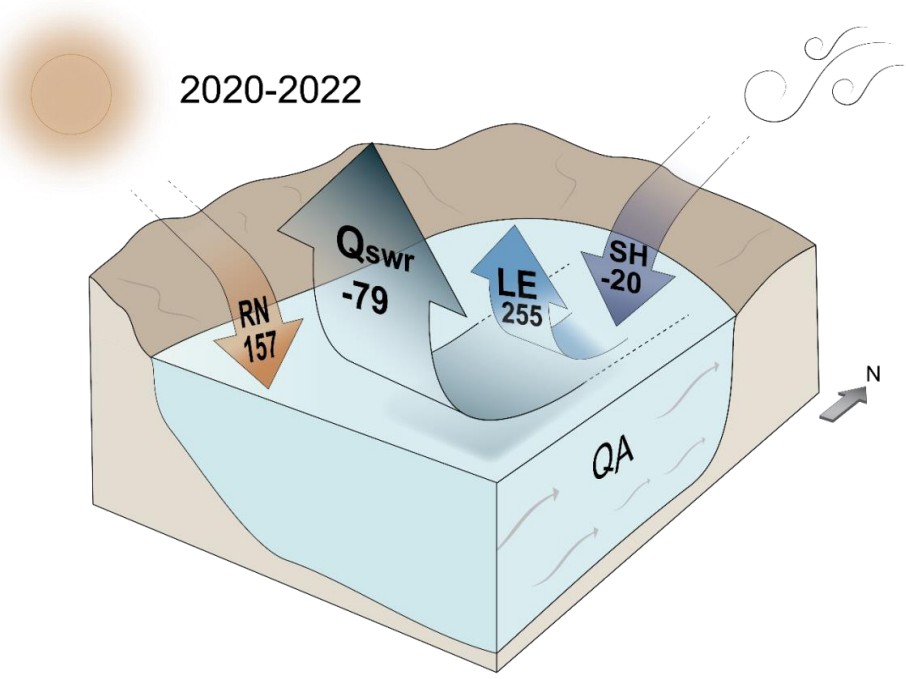

**Graphical abstract: Schematic representation of the mean energy partitioning at the Gulf of Eilat during the 2-year observation period (values are given W m$^{-2}$). The mean annual surface energy loss is represented by $Q_{SWR}$, which is the sum of the net radiation (RN), latent (LE) and sensible (SH) heat. The heat deficit is balanced by heat advection $Q_A$ through the northbound currents.**

**1. Introduction**

Research of air-sea exchange of heat, gas, and momentum through surface fluxes, provides valuable insights into the physical and chemical processes of marine environments that sustain diverse ecosystems, such as coral reefs. Accurate quantification of surface heat exchange is crucial for understanding thermoregulation of the shallow water environment and basin scale thermo-haline dynamics.

The principle of energy conservation provides a comprehensive framework for quantifying thermoregulation in a marine environment; it is captured by the following equation:

$$RN = LE + SH + G + Q_A, \tag{1}$$

The net radiation ($RN$) is the aggregation of the four components of the radiation budget, which are the downwelling ($DSW$) and upwelling ($USW$) shortwave radiation and are related through the albedo term of the

water, and downwelling ($DLW$) and upwelling ($ULW$) longwave radiation. Latent heat flux ($LE$) and sensible heat flux ($SH$) can be accurately measured, G refers to the change in heat storage in the water column. Net horizontal advection of heat in the water due to currents and tides is represented by $Q_A$. Therefore, in areas where reverse estuarine circulation occurs and where fluvial discharge and rain contribution are negligible, air-water interaction is directly connected to the general circulation through $Q_A$.



In recent decades, the measurement of the energy exchange components in marine and lacustrine environments has undergone significant advancements, particularly by using Eddy Covariance (EC) towers that offer direct and accurate measurements of the turbulent fluxes of $LE$ and $SH$ (Lensky et al., 2018; Mor et al., 2018; Hamdani et al., 2018; McGowan et al., 2019; Rey-Sánchez et al., 2017; Pérez et al., 2020; Tau et al., 2022). These fluxes are commonly estimated by bulk formulae methods due to the accessibility of the method.

The bulk formulae method is a widely adopted approach in estimating surface energy fluxes due to its reliance on easily obtainable variables, namely: sea surface temperature, air temperature, and a 2D wind measurement at 10 meters. The bulk formulae method calculates the evaporation flux by multiplying the wind speed, specific humidity difference, and vapor transfer coefficient (Kondo, 1975). This approach for estimating surface energy fluxes relies heavily on the parametrization of the vapor transfer coefficient, which involves several assumptions

and empirical relations for calculating the Monin-Obukhov length and the friction velocity (Fairall et al., 1996). Both can vary significantly in coastal regions according to the parametrization method (Bardal et al., 2018). While projects such as TOGA-COARE (Fairall et al., 1996) have aimed to reduce uncertainties in the vapor transfer coefficient, their findings are primarily applicable to the tropical open ocean, where an unstable marine atmospheric boundary layer (MABL) is persistent. However, in stable atmospheric boundary layer (ABL)

conditions that are common in sea areas where flow originates from land, like the northern Red Sea (as well as the Persian Gulf and the main body of the Red Sea), and coastal upwelling areas, large uncertainties remain in momentum and scalar flux estimation (Edson et al., 2007). These shortcomings indicate that discrepancies between the EC method and bulk formulae may originate from the vapor transfer coefficient parametrization. Thus, to overcome these uncertainties a comparison to direct measurements is needed. Apart from the effect on

each air-sea flux, the uncertainties are propagated into the energy balance of the system and therefore there is no energy closure, and the energy balance closure problem is unresolved (Yu, 2018).

In coastal regions, synoptic scale variability alternates between regional mean flow and secondary circulation, impacting the intensity, duration, orientation, and travel distance of cross land-sea winds (Allouche et al., 2023). Processes spanning from diurnal to seasonal cycles, involving local micrometeorological parameters and

variability due to synoptic scale forcing, play a crucial role in influencing the energy balance partitioning and thermoregulation of shallow marine environments. Synoptic scale forcing, which induces changes in humidity, wind speed, and air temperature, can rapidly influence water temperature and the overall energy balance (Abir et al., 2022; Genin et al., 2020; MacKellar and McGowan, 2010; Papadopoulos et al., 2013). These findings underscore the importance of understanding how the energy balance of marine environments behaves on a diurnal

to seasonal cycle in response to local and regional micrometeorological factors and irregular synoptic scale events. Recognizing these mechanisms as a research priority is crucial, as they are applicable to other regions experiencing strong land-sea winds (Azorin-Molina and Chen, 2009).

This paper presents a comprehensive analysis of continuous two years of EC measurements of heat and water vapor over the northern Gulf of Eilat (GoE). This data allows us to accurately characterize for the first time the

diurnal and seasonal cycles of energy balance partitioning in an arid semi-enclosed sea. The GoE is not only a model for other arid semi-enclosed seas; it has been proposed as a natural refugium for corals, as corals within the Gulf show unique thermal resistance, able to survive bleaching in extremely high temperatures (Fine et al., 2013; Abir et al., 2022). Thus, The Gulf is a critically important ecological and economic zone for Israel, Egypt, Saudi-Arabia and Jordan, and understanding the processes controlling its heat balance is of wide interest.



The results show discrepancies to previous studies' estimates by bulk formulae; both in mean rate and seasonal trend. Our analysis of the origin of the gap between the bulk formulae method to our direct measurements highlights the features that should be improved in the bulk formulae approach when applied in semi-enclosed seas. We also analyze synoptic scale events as a methodological case study; to demonstrate how synoptic scale events affect the energy balance. By characterizing air-sea interaction processes we gain insight into the

thermoregulation processes of semi-enclosed seas and thermo-haline circulation processes.

The structure of the paper is as follows: In Sect. 2 we describe the study region-specific setting and previous air-sea fluxes reports, the study methods, and data. In Sect. 3 we present the results of the study in the context of diurnal to seasonal cycles, comparison to bulk formula, and synoptic scale case study. In Sect. 4 we discuss the effect of the air-sea fluxes on thermoregulation processes, estimates of the advection flux, bulk formulae deviation

from EC results, and synoptic scale events as preconditioning to basin winter vertical mixing of the water column.

### 2. Study region, methods, and data

### 2.1. Study region

The study was conducted over the north-western shore of the GoE, Israel (Fig. 1a,b), a narrow (<25 km), deep (maximal depth 1800 m) sea semi-enclosed by a hyper-arid desert. The GoE is connected to the Red Sea through

the Straits of Tiran in the south. Local winds are channeled by the Arava rift valley, bringing (most of the time) dry terrestrial northeasterly wind (along gulf component) to the northern GoE (Fig. 1e). During winter, less common southerlies affect the region. These are channeled by the GoE basin, traveling a longer distance over the sea compared to the prevailing northerly winds (Afargan and Gildor, 2015).



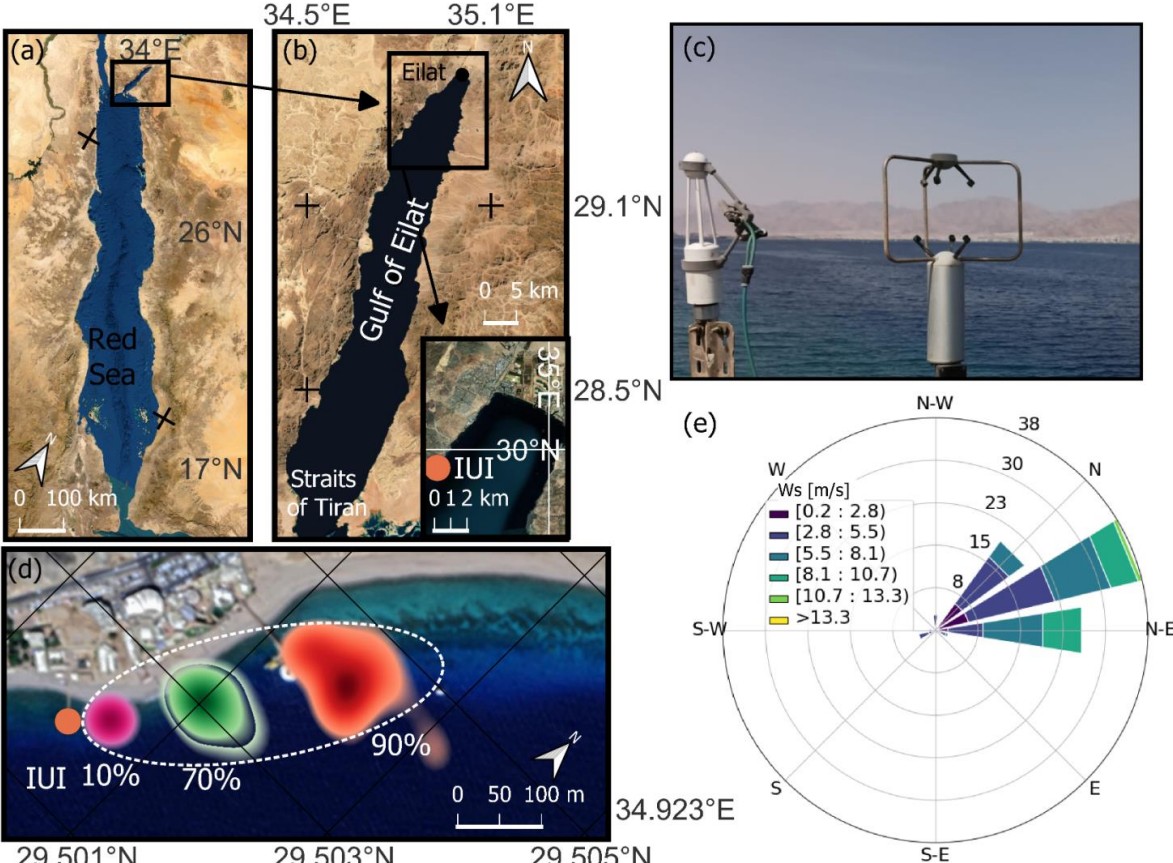

**Figure 1: Location map and site description.** Satellite images of (a) the Red Sea, (b) the Gulf of Eilat (Aqaba), and (b insert) the eddy covariance station location at the IUI. (c) The eddy covariance instruments, IRGA, and 3D wind anemometer. The footprint heatmap is presented in (d); pink=10%, green=70%, and red=90% represent the percentile of the distances from which the data was collected by the EC, and the white dashed line is a schematic representation of the footprint area. (e) Wind rose diagram for the entire observation period; legend values are wind speed in m s⁻¹ (September 2020- September 2022), and centric contours are percentile from the observations. Image ©: (a-b)- Data SIO, NOAA, U.S. Navy, NGA, GEBCO image © Landsat/ Copernicus, (b insert)- Data SIO, NOAA, U.S. Navy, NGA, GEBCO image © 2022 TerraMetrics, © 2022 CNES / Airbus, © 2022 Maxar Technologies, (d)- image ©2022 CNES/Airbus.

During summer (June, July, August hereafter JJA) the prevailing atmospheric synoptic circulation is a low-pressure system centered north-west of Israel (Persian Trough). During winter the common synoptic systems are Cyprus lows, Red Sea Trough (RST), and Siberian High. The Red Sea Trough (RST) is characterized by a region of surface low-pressure extending from the Red Sea to the coast of Turkey; it is common during transitional seasons and winter (Dayan et al., 2012). During winter (December, January, and February hereafter DJF) anti-cyclonic synoptic flow (intensified Arabian High an extension of the Siberian High) associated with surface high-pressure systems, can cause strong, cold, and dry westerly winds at the Red Sea. These anti-cyclonic flows result in extreme heat loss events through the latent and sensible heat fluxes (stronger than -400 W m⁻²) (Menezes et al.,



2019). In contrast, low-pressure systems are associated with lower heat loss values (values from -100 to -50 W m$^{-2}$), as these events are accompanied by warm and humid southerly winds (Papadopoulos et al., 2013). However, research on extreme heat loss events has been done on coarse satellite data (~0.5º), therefore an accurate representation of the GoE is lacking due to its small dimensions and indirect flux estimation (conducted on the OAFlux and MERRA-2 dataset).


Short periods of EC measurements conducted at the GoE (Rey-Sánchez et al., 2017; Abir et al., 2022) during summer measured an evaporation rate of >10.3 mm day$^{-1}$. Thus, showing discrepancies between the evaporation rates estimated by bulk formulae during summer (~3 mm day$^{-1}$) and maximum evaporation during winter (11 mm day$^{-1}$). These bulk formulae explained the higher winter evaporation by the ABL instability due to warmer surface


water than the overlying air (range of bulk formulae estimation: [1.6,3.65] m year$^{-1}$ (Ben-Sasson et al., 2009; Eshel and Heavens, 2007; Sofianos et al., 2002; Assaf and Kessler, 1976)). These findings highlighted the need for implementation of accurate and direct measurement methods of the turbulent fluxes, to better understand the complex energy balance dynamics in the GoE. Similar characteristics probably exist in other desert-enclosed basins, such as the Red Sea or the Arabian Gulf.


The general circulation model of the GoE which was originally proposed by Klinker et al. (1976), was characterized as reverse estuarine circulation and was the widely accepted model. The model postulated that buoyancy circulation driven by heat loss and evaporation is the primary driving force behind the GoE circulation. Biton & Gildor (2011b) proposed a seasonally varying model, where the maximal exchange flux between the northern Red Sea and the GoE occurs during the re-stratification season (April to August). It is, driven by density


differences between the basins, while atmospheric fluxes counteract this exchange flow. They attributed the observed warming of the surface primarily to the advection of warm water from the northern Red Sea, with a smaller contribution from surface heating.

Since the winter of 2011/2, the vertical winter basin overturning (vertical mixing) did not exceed a depth of ~500 meters until the winter of 2021/2 which exceeded 700 meters (Shaked and Genin, 2022). The depth of the mixing


is strongly affecting the nutrients budget of the shallow water at the GoE, by transporting deeper water enriched with nutrient to the surface.

The alongshore currents in the northwestern terminus shore where the EC was located typically flow from north to south in response to tides and winds (Berman et al., 2000, 2003). Daily shallow water temperature at the GoE varies from ~21.5 °C (minimum 20.5 °C) during winter to ~27.5 °C (maximum 30.5 °C) during summer (based


on measurements from 2007 to 2021 conducted by Israel's National Monitoring Program (NMP)). The average annual open sea SST increased by an average rate of more than 0.5 °C decade$^{-1}$ since 2004 (Shaked and Genin, 2022).

### 2.2. Methods

#### 2.2.1. Instruments


##### 2.2.1.1. Eddy Covariance tower

The eddy covariance system consisted of an open path CO2/H2O infrared gas analyzer (IRGA; model LI 7500, LI-COR, Inc., USA) coupled with a three-dimensional ultrasonic anemometer (R.M. YOUNG 81000), both recorded at 20 Hz using a CR1000X data logger. The instruments were positioned 45 cm apart on the same vertical level. The station (Fig. 1c) was mounted on a ~7 m high mast above the mean sea surface, ~36 m offshore,


positioned at the end of the wharf (29.510748°N, 34.917669°E) of the Interuniversity Institute for Marine Sciences



(IUI). Solar and maritime radiation exchanges were measured by a CNR1/CNR4 net radiometer (Kipp & Zonen B.V., The Netherlands) mounted on the south side of the wharf of the IUI station extending ~1.5 m out from the wharf, and 1.5 m above the water surface. Air temperature and relative humidity were measured by HC2S3 probes (Campbell Scientific, USA). Water skin temperature was measured by an SI-4H1 infrared radiometer (Apogee

Instruments, Inc.). Water temperature (measured at a water depth of ~1 m) at IUI and also relative humidity, air temperature, $DSW$, and wind measurements for gap-filling purposes were obtained through the NMP at the GoE (http://www.meteo-tech.co.il/eilat-yam/eilat_en.asp). The LI-COR 7500 open path gas analyzer and net radiometer were regularly washed to ensure the sensors were free of salt and dust.

### 2.2.1.2. High-frequency hydrographic data

We used an ocean mooring equipped with a Del Mar Oceanographic's WireWalker, hereinafter WireWalker. The WireWalker is purely mechanical and profiles continuously, with wave energy driving the buoyant profiler downward. On reaching the deepest user-specified sampling depth, it free-ascends along its suspension cable (aka profiling wire), nearly completely decoupled from mooring motion. The WireWalker was equipped with an RBRconcerto CTD. The RBRconcerto CTD was programmed to work in Directional mode: with fast sampling (8

Hz) in the ascending direction and a slow sampling rate (1 Hz) while descending. In this work, we used only upcast data. The RBRconcerto CTD measured the temperature and salinity from 3 m down to 150 m between 10/11/2021 06:00:00 and 23/11/2021 12:14:55, 08/02/2022 6:22:35 - 03/03/2022 8:34:35, and 06/07/2022 8:00:00 and 27/07/2022 10:43:52.

### 2.2.2. Footprint and data quality

EC measurement footprints were calculated using the EddyPro© program (LI-COR Biosciences, 2021) (Fig. 1d). To minimize data "contamination" of the footprint a wind direction filter was applied to exclude wind from land (excluded wind directions: 225°-360°). The water surface is not static due to water motion (currents, waves, tides), however, we can regard it as a static surface, since the water motion velocity is two orders of magnitude smaller than wind velocity. Maximum tidal changes in the GoE are small and may offset the 90% footprint distance by

only a few tens of meters, i.e. 622 to 700 m (~ 10% change), and are therefore considered not to greatly affect energy flux measurements. In addition, low-quality data were excluded, using the Foken flag method (Foken et al., 2004) (data categorized as Foken flag=2 was excluded), and post-processing manual spike removal was conducted so that latent heat ($LE$) and sensible heat ($SH$) fluxes were restricted between -50 to 1100 W m$^{-2}$ and $\pm$ 250 W m$^{-2}$ respectively. The resulting percentage recovery of 30-minute mean flux data was 45%/46% ($LE/SH$)

before gap-filling which is typical of the recovery rate of shoreline-mounted EC stations (Gutiérrez-Loza and Ocampo-Torres, 2016). During April- June 2022 the EC station recorded at a frequency of 4 Hz due to an error, the reader is referred to Figure S3 in Supporting Information 1 (SI1) for an analysis of the uncertainty in the fluxes due to this error, in which the mean difference for the latent and sensible heat is of 30 W m$^{-2}$ and 3 W m$^{-2}$ (respectively), between 4 and 20 Hz measurements.

### 205 2.2.3. Post-processing

This subsection describes the calculation of additional meteorological parameters relevant to the energy balance analysis, and the assumptions and methodologies of the calculation of the energy balance equations (Eq. (1)).

Half-hourly energy fluxes (Abir et al., 2023) were obtained using the EddyPro© program (LI-COR Biosciences, 2021). The program was run in express mode which includes defaulted filters and corrections - wind component

double rotation, block averaging, and spectral corrections (EddyPro® 7 Software: Express default settings, 2023).



The contribution of rain to the energy balance is negligible, as the study site is located in a hyper-arid desert (annual average precipitation between 1990-2020 is 23 mm (Eilat annual average precipitation, 2023)). The heat exchange across the seabed is also considered to be negligible in this study due to its typical values being usually a minor contributor to the energy balance equation (Pivato et al., 2018; Shalev et al., 2013). $Q_A$ is hard to measure
in the GoE since geopolitical constraints prevent across-gulf measurements of currents and temperatures, and in this paper is inferred as a residual from the surface energy fluxes ($RN, LE,$ and $SH$) and the change in heat storage. When the change in heat storage measurement is not available, then sea water residual will be reported ($Q_{SWR}$) which is the net surface exchange ($Q_{SWR} = RN - LE - SH$) (McGowan et al., 2010).

Change in heat storage in the water column ($G$) was obtained by first calculating the heat storage as followed
from the WireWalker CTD measurements:

$$Hg = \rho \cdot C_p \cdot \overline{T} \cdot \Delta Z, \tag{2}$$

Where $Hg$ is the total heat of the profile water column in J m$^{-2}$, $\rho$ is the seawater density, $C_p$ is the heat capacity at a constant pressure of seawater ($\rho \cdot C_p = 4.1 \cdot 10^6$ [J m$^{-3}$ °C$^{-1}$]), and $\Delta Z$ in meters is the 1-meter water layer thickness. Then the heat storage was differentiated in time for the change in heat storage:

$$G = \frac{\Delta Hg}{\Delta t}, \tag{3}$$

Where $G$ is the change in heat storage in W m$^{-2}$, $\Delta Hg$ is the change in total heat, and the $\Delta t$ in second (for the full description of the calculation the reader is referred to the SI1).

The vapor pressure difference between the water surface and the overlying air is given by:

$$\Delta e = \beta \cdot e_{sat}(T_w) - \frac{RH}{100} \cdot e_{sat}(T_a) = e_w - e_a, \tag{4}$$

where $Tw$ and $Ta$ are the water and air temperature, respectively, and are given in °C, $e_{sat}(T)$ in mbar is the saturation vapor pressure by the Magnus-Tetens formula, $\beta$ is water activity set to be 0.98 which is a typical value for seawater (Sverdrup et al., 1942), $RH$ is the relative humidity, $e_w$ and $e_a$ are the water and air vapor pressure in mbar (respectively), and $\Delta e$ is the vapor pressure difference in mbar.

### 2.2.4.    Gap-filling

Gaps (gaps duration ~6 h) in the relative humidity ($RH$), wind speed ($W_s$), water temperature ($Tw$), and air temperature ($Ta$), where data was missing due to station quarterly maintenance. These gaps were filled first by replacing the missing data with the data from an adjacent sensor from the NMP. Following that, a bi-linear, periodic trended interpolation (Morin et al., 2014) was used for small gaps.

Due to the malfunction of the downward-facing pyrgeometers of the CNR1 (September 2020-April 2022), the
$ULW$ was calculated in W m$^{-2}$ from the water temperature measurement according to the Stefan-Boltzmann law:

$$ULW = \varepsilon \cdot \sigma \cdot Tw^4, \tag{5}$$

Where $\varepsilon$ is the emissivity ($\varepsilon = 0.985$), $\sigma$ is the Stefan-Boltzmann constant ($\sigma = 5.6697 \times 10^{-8}$ W m$^{-2}$ K$^{-4}$) and $Tw$ is the water temperature in °K.

The $DLW$ was calculated based on Bignami et al. (1995); cloud coverage effect on the calculation was determined
to be negligible (Ben-Sasson et al., 2009):

$$DLW = \sigma \cdot Ta^4 \cdot (0.65 + 0.00535 \cdot e_a), \tag{6}$$



Equations (5) and (6) were used to gap fill during the period where the station measured in 4 Hz mode which interfered with the data collection of the CNR4, and general gap filling.

The $USW$ was calculated as a fracture of the downwelling shortwave radiation (Paynes, 1972):

$$USW = 0.065 \cdot DSW,\tag{7}$$


The $DSW$ was gap filled by taking the measurement from the adjacent NMP sensor and Eq. (7) was used to fill gaps in the $USW$. The difference between the calculated net radiation mean and the measured net radiation mean was 50 W m$^{-2}$, which is 20% out of the measured mean (during a month where there were no gaps in the net radiometer data).

To fill gaps in $LE$ and $SH$ an Artificial Neural Network (ANN) algorithm was utilized (Mahabbati et al., 2021). The algorithm was implemented with Python sklearn.neural_network MLPRegressor package when the hyperparameters (learning rate, hidden layer, and maximum iteration number are specified in the SI1 Figure S1, 2) were chosen by a grid search cross-validation algorithm (for the evaluation of the ANN performance refer to Figures S1, 2 in the SI1). Following the gap-filling procedure, the resulting percentage recovery of 30-minute

mean flux data was >99%.

### 2.2.5. Bulk formula

The bulk formula used to compare air-sea fluxes at the GoE is the COARE3.6 algorithm (Fairall et al., 1996). We used the NMP measurements as input data for the algorithm, hence, this analysis is comparable to previous works that utilized this data and for future reference. The Bulk Richardson number ($Ri_B$) which is used by the bulk

formula to indirectly parameterize the stability parameter ($\zeta$) (for more details see section 'Calculating atmospheric stability' in the SI) is given by:

$$Ri_B = \left(\frac{g}{Ta+273.16}\right)\left[(\Theta - \Theta_0 - f(Ta,\Delta q)) \cdot \frac{z}{(Ws)^2}\right],\tag{8}$$

Where $g$ (m s$^{-2}$) is the gravity acceleration, $\Theta$ and $\Theta_0$ (°K) are the potential temperature calculated from $Ta$ and $Tw$ respectively, $z$ (m) is the measurement height of the wind speed, $Ws$ is the wind speed (m s$^{-1}$), and

$f(Ta,\Delta q)$ is a correction term for the potential temperature difference due to the air lapse rate and cool skin effect ($\Delta q$ is the specific humidity difference between the water surface and the air in g Kg$^{-1}$). To compare the bulk formula vapor transfer coefficient ($C_e$) to the EC measurements, we calculate $C_e$ from the EC $LE$ flux:

$$C_e = \frac{LE}{Ws \cdot \Delta q \cdot \rho_a \cdot L_v},\tag{9}$$

Where $\rho_a$ (Kg m$^{-3}$) is the air density and $L_v$ (J g$^{-1}$) is the latent heat of vaporization of water, both calculated

following Fairall et al. (1996).



## 3. Results

### 3.1. Micrometeorology and water temperature

The 2-year (September 2020-September 2022) measured local micrometeorology and their mean diurnal cycles per season are presented in Fig. 2 and Fig. 3, respectively. The mean air temperature and relative humidity during the observation period were 25.9 °C and 42 % (respectively).






**Figure 2: Diurnal (LT) and seasonal variations (y and x axes, accordingly) of the following parameters, measured during September 2020-September 2022: (a) Air temperature ($Ta$), (b) relative humidity ($RH$), (c) water temperature ($Tw$), (d) water-air temperature difference ($\Delta T$), (e) wind speed ($Ws$), (f) vapor pressure difference ($\Delta e$), (g) $\Delta e \times Ws$, (h) net radiation ($RN$), (i) latent heat flux ($LE$), (j) sensible heat flux ($SH$), (k) sea-water residual ($Q_{SWR}$), (l) daily sea water residual change ($\langle \Delta Q_{SWR} \rangle$).**

The air temperature varied between the winter seasonal mean value of 18.7 °C to the summer seasonal mean value of 32.1 °C. Relative humidity seasonal mean varied between a value of 47 % during autumn (September, October, November hereafter SON) and winter (DJF) to a value of 39 % during summer (JJA) and spring (March, April, May hereafter MAM) (Appendix A). The water temperature mean value was measured to be 24.7 °C, $Tw$ was characterized by a low mean water temperature value of ~23 °C during winter and spring which then shifts to summer and autumn with higher mean values of ~26 °C. Consequently, the water at the GoE is warmer than the overlying air from November to March (Fig. 3f and Figure S4 in SI1). The northeasterly (mean direction 23º) dominating winds have a mean seasonal wind speed of ~4.9 m s⁻¹ observed through spring, summer, and autumn. The wind speed reaches the minimum value of 4.1 m s⁻¹ during the winter (minimum out of the mean value per season). The seasonal mean diurnal cycle of wind speed (Fig. 3b and Figure S4 in SI1) can be differentiated into two common cycles: Double peak and the bell-shaped curve. Both types, start to increase from the lower nighttime values, at ~06:00 LT reaching the first daily maximum of >6 m s⁻¹ at ~10:30 LT, eventually, at ~15:00 LT, the wind speed decreases to <5 m s⁻¹ in winter, and at 17:00-17:30 LT during the rest of the year. The diurnal variations of $\Delta e$ (Eq. (4), Fig. 3d) were characterized by daily mean minimum values occurring at ~09:00 LT (14 mbar) and the daily mean maximum values at ~18:30 LT (18 mbar). Additionally, from April-October the mean daily variations of $\Delta e$, are significantly more pronounced, concurrently with the water-air temperature difference ($\Delta T$) and the $RH$ daily variations. In winter the highest value of seasonal mean $\Delta e$ is obtained (17.3 mbar) due to the positive $\Delta T$ (Fig. 3f) however, during the summer higher mean maximum daily values (26.3 mbar) are obtained due to the strong diurnal variations.




**Figure 3: Seasonal mean diurnal (LT) cycle of micrometeorological and flux variables (Winter=December-January-February=DJF, Spring=March-April-May=MAM, Summer=June-July-August=JJA, Autumn=September-October-November=SON), and in the shaded area the +/- 1 SD span. (a) Energy fluxes, (b) wind speed ($Ws$), (c) relative humidity ($RH$), (d) vapor pressure difference ($\Delta e$), (e) vapor pressure difference product with wind speed ($Ws \times \Delta e$), (f) water-air temperature difference ($\Delta T$).**

### 3.2. Surface heat flux

The 2-year (September 2020-September 2022) measured surface fluxes and their mean diurnal cycles per season are presented in Fig. 2 and Fig. 3, respectively.



### 3.2.1. Net radiation

315    The observation period mean $RN$ value was 157 W m$^{-2}$, where positive values represent flux of energy from the atmosphere into the water surface. Net radiation mean diurnal cycle was of the characteristic bell-shaped and

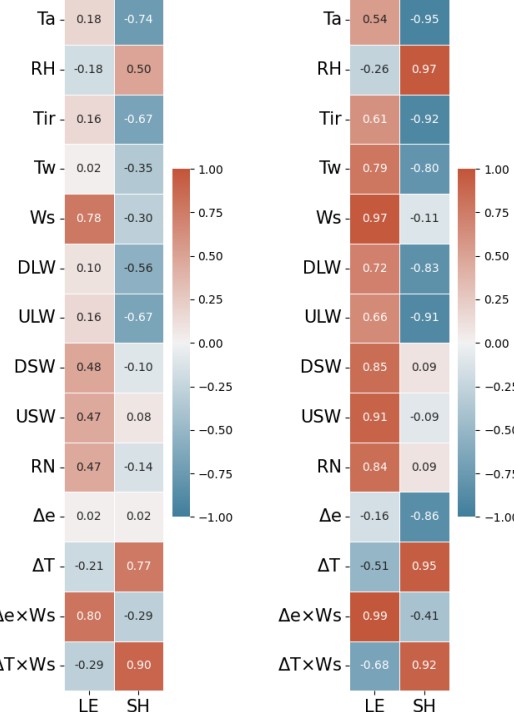

seasonal mean values varied between 55 W m$^{-2}$ in winter to 252 W m$^{-2}$ during summer (Fig. 3a). Nighttime net radiation values were negative, with stronger $ULW$ than the $DSW$ (Fig. 3a), and with positive $RN$ values during the daytime. The daily mean maximum temperature of $Tw$ lags behind the mean daily maximum $RN$ by 2.5 h,

320    while the daily mean maximum $Ta$ lags by 5.5 h.



**Figure 4: 30-minute data (left) and daily mean data (right) Pearson's correlation matrix of the sensible and latent heat fluxes with micrometeorological and radiation flux variables.**

### 3.2.2. Latent heat

The mean $LE$ flux during the observation period was 255 W m$^{-2}$ (3.22 m year$^{-1}$ of evaporation). Latent heat flux seasonal mean varied between the minimum values during winter (232 W m$^{-2}$) to the maximum during summer (276 W m$^{-2}$). As described in Sect. 3.1 for the $Ws$ mean diurnal cycle, the $LE$ flux mean diurnal cycle is categorized into two cycles; double peak and bell-shaped curve, where from mid-spring (April/May) to mid-autumn (September/October) there was a clear double peak in the diurnal $LE$ cycle. The first peak coincides with the daily maximum wind speed (~10:00 LT) whereas, the second peak which was the absolute daily maximum (mean max value: 462 W m$^{-2}$) coincides with the product of the $Ws$ and the $\Delta e$ (Fig. 3e) at 15:30 LT. The product of the $Ws$ and the $\Delta e$ as prescribed by the bulk formulae approach has the highest correlation to the $LE$ flux (0.99, Fig. 4 right). However, at the GoE the high correlation originated mostly from the correlation to the wind speed profile of 0.97 (Fig. 4 right). The high correlation was visible in the temporal variations in both diurnal and seasonal plots (Fig. 3,4).

### 3.2.3. Sensible heat

The $SH$ flux mean value through the observation period was -20 W m$^{-2}$. Concurring with the seasonal air-sea temperature difference cycle, the $SH$ flux on average transfers heat from the water to the atmosphere during winter (winter mean values ~ 19 W m$^{-2}$), and continues outside of winter to September and May (Fig. 3a and Figure S4 in SI1). Spring to autumn mean seasonal value of $SH$ flux ~(-32) W m$^{-2}$ corresponds to the relatively cool sea surface in a hot desert surrounding. Similar to the $LE$ flux, the $SH$ flux as well is highly correlated to the product of the $Ws$ and a gradient; air–sea temperature difference (Pearson r=0.92, Fig. 4 right).



### 3.2.4.    Evaporation estimations – direct EC measurements vs. bulk formula

Figure 5 presents the bulk formula calculation with the COARE3.6 algorithm of ABL meteorology parameters and the latent and sensible heat fluxes throughout the 2-year observation period compared to the EC-obtained variables.

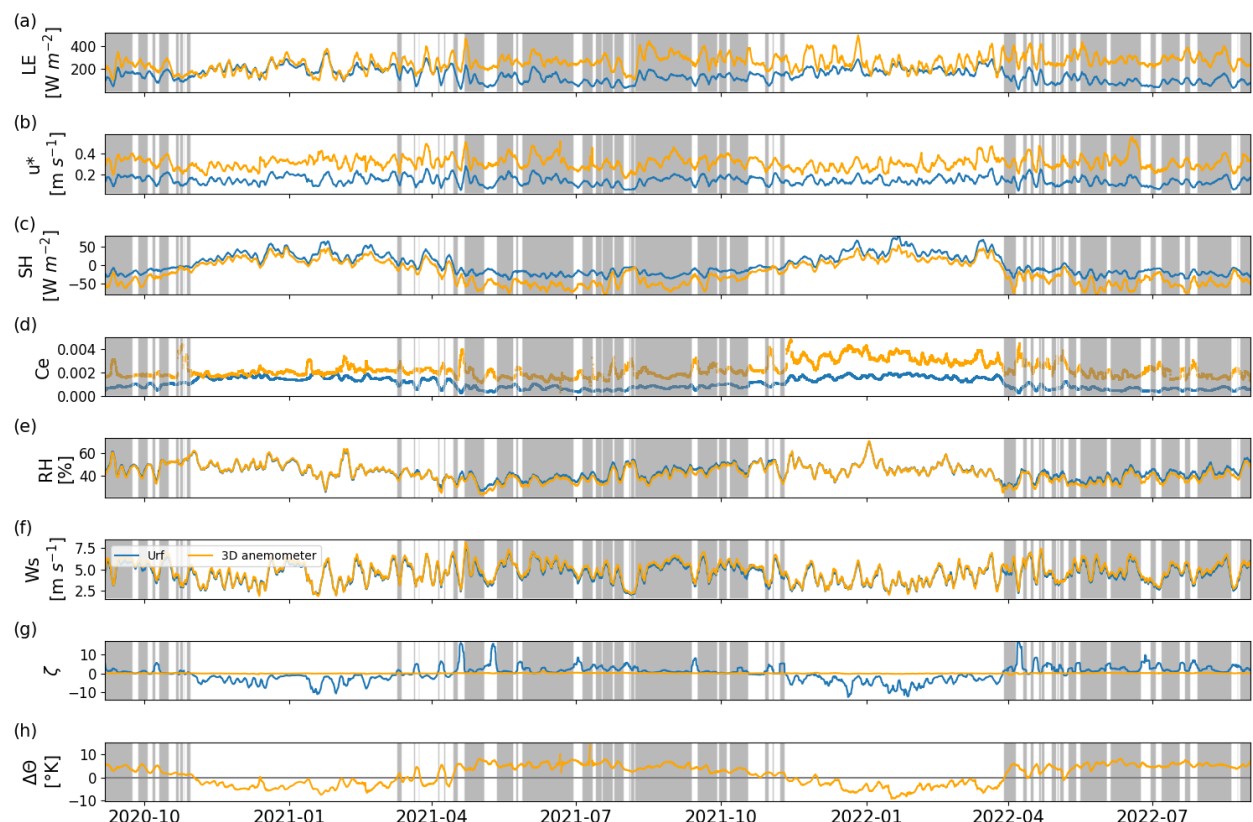

**Figure 5: Comparison between the 3-day (LT) rolling average EC-measured sensible and latent heat fluxes and meteorological parameters (orange) and the bulk formula calculation using TOGA-COARE 3.6 algorithm (blue)**
**computed with the NMP micrometeorological data. Shaded in gray are instances where the TOGA-COARE 3.6 algorithm bulk Richardson number is within the range of 0-0.2 indicating stable conditions. The wind speed panel compares the measurements of the 3D sonic anemometer and the interpolated referenced height (7 m) of the TOGA-COARE 3.6 algorithm. (a) Latent heat ($LE$), (b) friction velocity ($u^*$), (c) sensible heat ($SH$), (d) vapor transfer coefficient (reversed calculated from the EC measurements Eq. (9)), (e) relative humidity ($RH$), (f) wind speed ($Ws$),**
**(g) stability ($\zeta$), (h) air and water potential temperature difference ($\Delta\theta$). Key variations between the EC method and the bulk formula are higher evaporation rates measured with the EC system during stable conditions, consistent higher friction velocity measured with the 3D sonic anemometer, and low variation within the stability parameter calculated with the EC system.**

The bulk formula for $LE$ and $SH$ at the GoE under-estimate (Table 1): During winter bulk formula mean (Fig.
5a,c respectively) values were the closest to the higher EC values. In Spring the difference between the two



methods increased, reaching a maximum during summer. The annual maximum of the seasonal mean of the EC measurements was during summer whereas the bulk formula's was during winter.

| | Winter | | Spring | | Summer | | Autumn | | Annual mean | |
|---|---|---|---|---|---|---|---|---|---|---|
| | *LE* | *SH* | *LE* | *SH* | *LE* | *SH* | *LE* | *SH* | *LE* | *SH* |
| EC (W m⁻²) | 232 | 19 | 271 | -25 | 276 | -49 | 242 | -22 | 255 | -20 |
| Bulk formula (W m⁻²) | 199 | 34 | 144 | -3 | 98 | -22 | 146 | -6 | 147 | 0.4 |


**Table 1: EC and bulk formula seasonal and annual mean $LE$ and $SH$ fluxes comparison.**

The measured 3D anemometer mean friction velocity (0.31 m s⁻¹) was ~2 times larger (Fig. 5b) than the bulk formula estimation from the 2D wind measurement (0.14 m s⁻¹). The ABL stability parameter ($\zeta$) (Fig. 5g) showed a pronounced seasonal variation reaching its minimum value during the winter of ~(-12) and maximum

value during the summer of ~17 with an overall mean value of -0.04. While $\zeta$ calculated by the EC system had a much less pronounced variation between the minimum and maximum values [-0.75,1.11], with a mean value of 0.06. The mean annual derived vapor transfer coefficient, $C_e$, from the EC $LE$ is 0.00229 (Eq. (9)) whereas the bulk formula $C_e$ is 0.00108 (Fig. 5d). However, the bulk formula seasonal means of $C_e$ varied between two orders of magnitude while the EC-derived $C_e$ varied in the scale of one order of a magnitude (seasonal mean range of

bulk formula $C_e$: [0.00064, 0.00165] and EC: [0.00186, 0.00276]). The difference between the daily mean values of the $SH$ flux calculated by the bulk formula and the EC approach is 20 W m⁻².

**3.2.5.    Sea water residual- net surface heat exchange**

The seawater heat gain or loss through the sea surface is defined as the seawater residual heat flux ($Q_{SWR}$), calculated as the sum of the measured surface fluxes $LE$ and $SH$, and $RN$, (i.e. net surface fluxes). The measured

$Q_{SWR}$ are presented in diurnal and seasonal diagram Fig. 2k-i, and as a diurnal mean per season in Fig. 3a.

$Q_{SWR}$ mean observation period value was -79 W m⁻², which lead to a net cooling of the surface water through the surface fluxes. Apart from summer, the seawater residual was negative, reaching a minimum seasonal mean value of -195 W m⁻² during winter and a maximum seasonal mean value of 27 W m⁻² in summer. The mean diurnal cycle of the seawater residual was positive through ~08:00-15:00 LT, then in the afternoon, it was negative for

the entire night and early morning. The change from the positive values of seawater residual to negative values in the afternoon coincides with the maximum of the mean daily cycle of the water temperature (Fig. 3 and Figure S4 in SI1).

***3.2.6.    The change in the stored heat and Sea water residual***

Figure 6 presents the available measured heat transfer components of the surface and water column heat exchange

during three campaigns and also the advection flux as a residual form of the energy balance equation (Eq. (1)). Examining data from three campaigns for calculating GoE heat storage in the top 145 m of surface water showed





that in the three campaigns, the deficit required to close the energy heat balance was 74, 88, and 175 W m$^{-2}$ during winter, summer, and autumn respectively. During the winter campaign, there was a net cooling of the upper mixed layer, 0-145 m, (water column temperature and salinity ranged between 21.4-22.1 °C and 40.34-40.85 PSU, respectively). The net cooling is equivalent to -83 W m$^{-2}$ in heat storage change ($G$). During summer there was net heating of the strongly stratified water (water column temperature and salinity ranged between 21.4-28.0 °C and 40.45-40.78 PSU, respectively), which was equivalent to a 95 W m$^{-2}$ in $G$. During both campaigns, $G$ did not equal $Q_{SWR}$. During autumn $G$ of the mixed top water column was negligible (5 W m$^{-2}$) and the largest deficit in the heat balance closure exists due to the strong surface cooling ($Q_{SWR}$=-170 W m$^{-2}$).





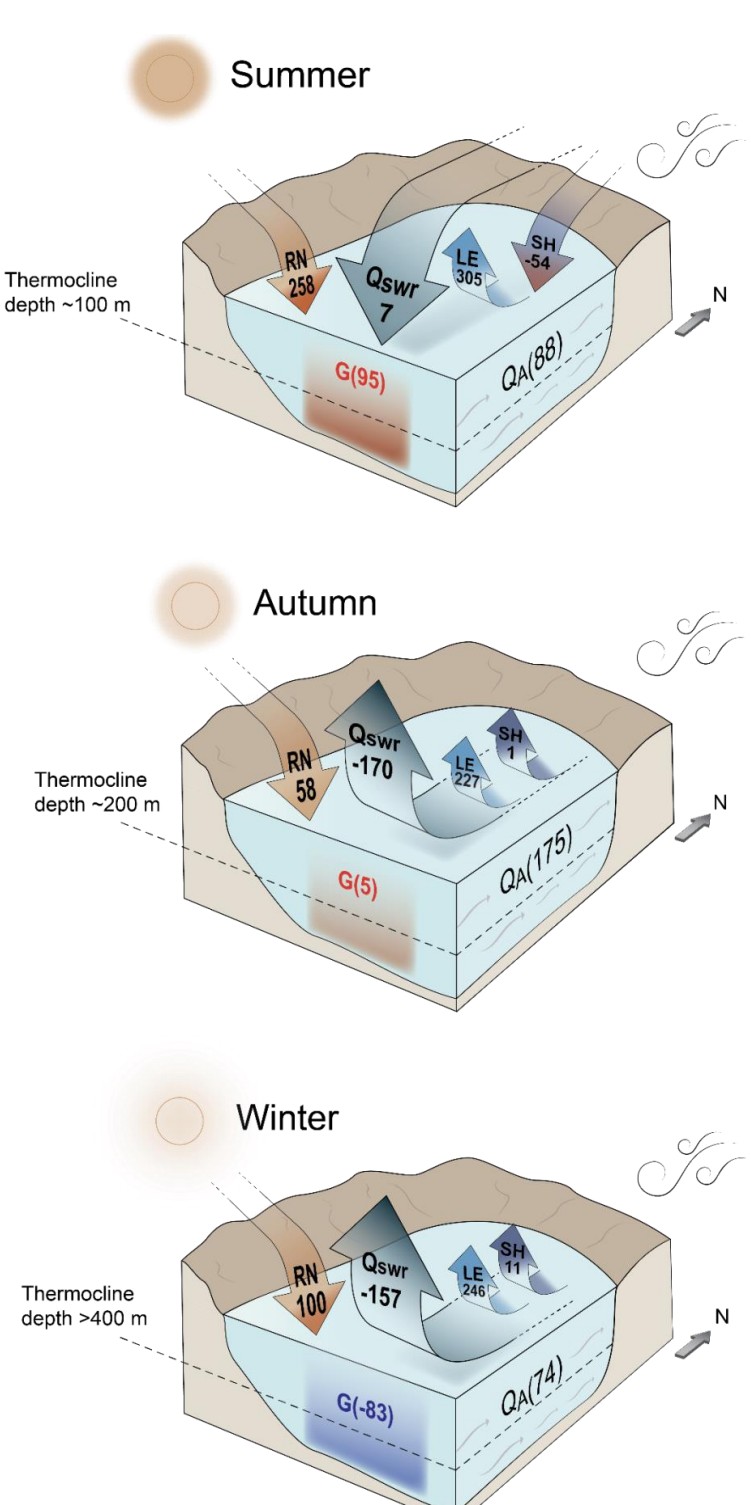



**Figure 6: Schematic representation of the mean energy partitioning at the GoE during the three campaigns when water temperature profiles are available (values are given W m$^{-2}$).**

### 3.3. Synoptic forced winter heat loss event

We examine here how irregular synoptic scale forcing events drive intensive heat loss from the water surface to the atmosphere. In the course of 3 days (25-27 December 2021) intensification of wind speed, and cooler and dry air mass forced high rates of heat loss from the sea surface to the atmosphere by $LE$ and $SH$.

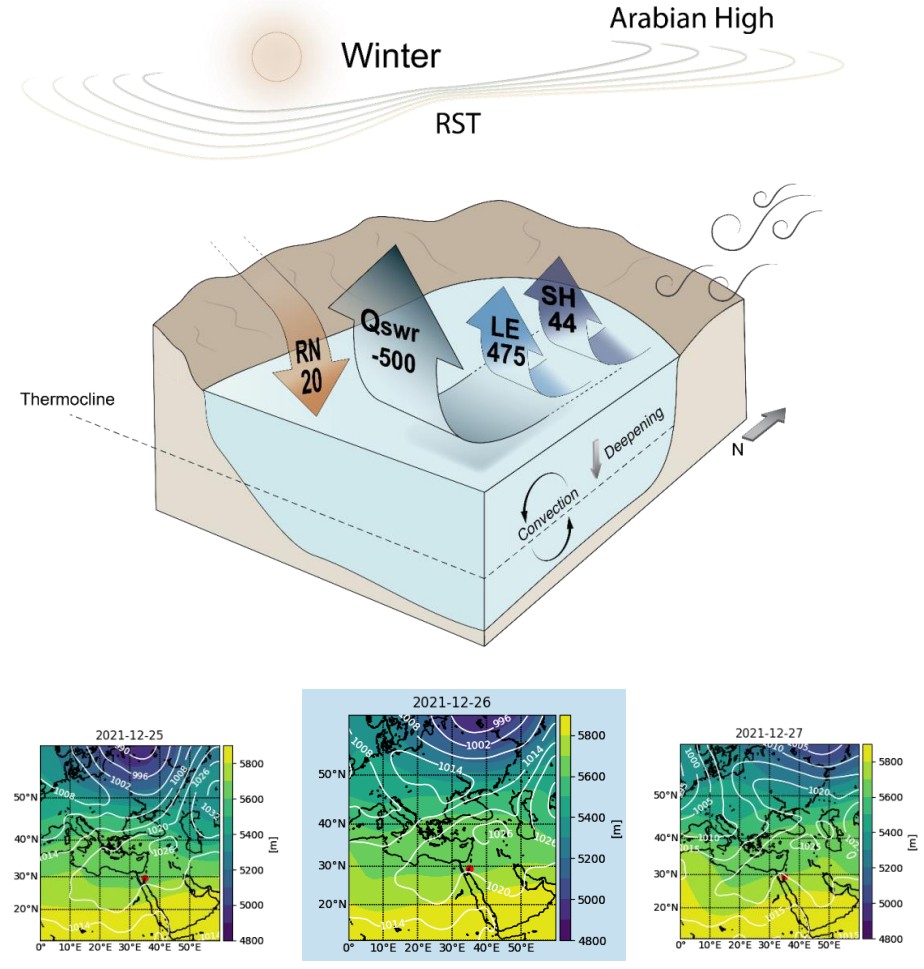

**Figure 7: Surface heat loss case study synoptic maps (bottom) and schematic representation of the mean energy**
**partitioning during the event (top). (Bottom) Average daily ERA5 reanalysis variables synoptic maps, sea level pressure (hPa) plotted in white in contour (Hersbach et al., 2023b), and the 500hPa geopotential height (m) plotted in shaded colors (Hersbach et al., 2023a). The event occurred between 25-27.12.2021 (shaded map marks the peak of the event) and involved the intensification of the Arabian High and the Red Sea Trough. (Top) Surface energy fluxes are given in W m$^{-2}$. The large heat loss is affecting the deepening of the thermocline and the winter annual vertical water column**
**mixing.**





Figure 7, presents the formation and decay of the RST (Alpert et al., 2004) surface-level synoptic state and intensification of the Arabian High, associated with a large surface-water heat loss event. The described synoptic pattern caused an increase in daily mean wind speeds, from 3.8 to 5.9 m s$^{-1}$ (Fig. 8a) relative to the season mean (DJF 2021-2). This mean increase manifested also in an increase of the minimum and maximum daily wind speed

values at the GoE for the 3 days. Surface wind direction at the GoE was northeasterly (23º from the north), with an intensification of the westward component of the wind between 800 hPa to 500 hPa (Figure S5e in the SI1) in the region. Additionally, during the event colder air masses in comparison to the sea temperature, were present in the GoE (the $\Delta T$ during the event was 1.5 °C larger than the season mean). The increase in $Ws$ and $\Delta T$ forced large heat loss from the surface water of the GoE (Fig. 8i), -270 W m$^{-2}$ stronger than the DJF mean which is 1.7

standard deviation (SD) of the season mean. The largest mean daily heat loss occurred on 26 December (mean daily value $Q_{SWR}$=-560 W m$^{-2}$) coinciding with the peak of the $Ws$ during the event (8.9 m s$^{-1}$). The surface heat loss is a result of strong fluxes of $LE$ and $SH$ (Fig. 8f,g, respectively), which increased from 247 to 475 W m$^{-2}$ (2 SD increase from the season mean) and 19 to 44 W m$^{-2}$ (1.25 SD increase from the season mean), respectively. Unlike this event where the high $Ws$ and mean values of $RH$ and lower $Ta$ (Fig. 8b,c respectively) resulted in

large heat loss from the surface of the GoE, on 9 January 2022 (shaded yellow area in Fig. 8k) even higher mean daily wind speed (7.2 m s$^{-1}$) were measured. However, the mean $LE$ flux measured only 314 W m$^{-2}$, and $SH$ decreased to 2 W m$^{-2}$. Contrary to the portrayed heat loss event, on 9 January; warmer and more humid southerly winds persisted (184º). These air mass characteristics resulted in a decrease in $\Delta e$ and $\Delta T$ ($\Delta e = 8.5 mbar$, ~9 mbar less than the season and the event mean value, $\Delta T = 0.8$ °C), which did not allow high rates of heat loss in

comparison to the seasonal mean. Overall, during the winter season of 2021/2, there were 7 days where the mean daily $Q_{SWR}$ was stronger than -400 W m$^{-2}$ in comparison to the previous winter of 2020/1 where only 2 days were recorded (mean winter $Q_{SWR}$= -195 W m$^{-2}$). The synoptic pattern of 5 out of 7 of the days in winter 2021/2 was classified as either RST or high to the east (Arabian High) (see Alpert et al. (2004) for the classification method). Out of the 7 days, 2 events that prolonged a single day can be identified; the first is a 2-day event and the latter is

the prescribed 3-day event, thus large heat loss occurred mostly through events rather than sporadic heat loss days.



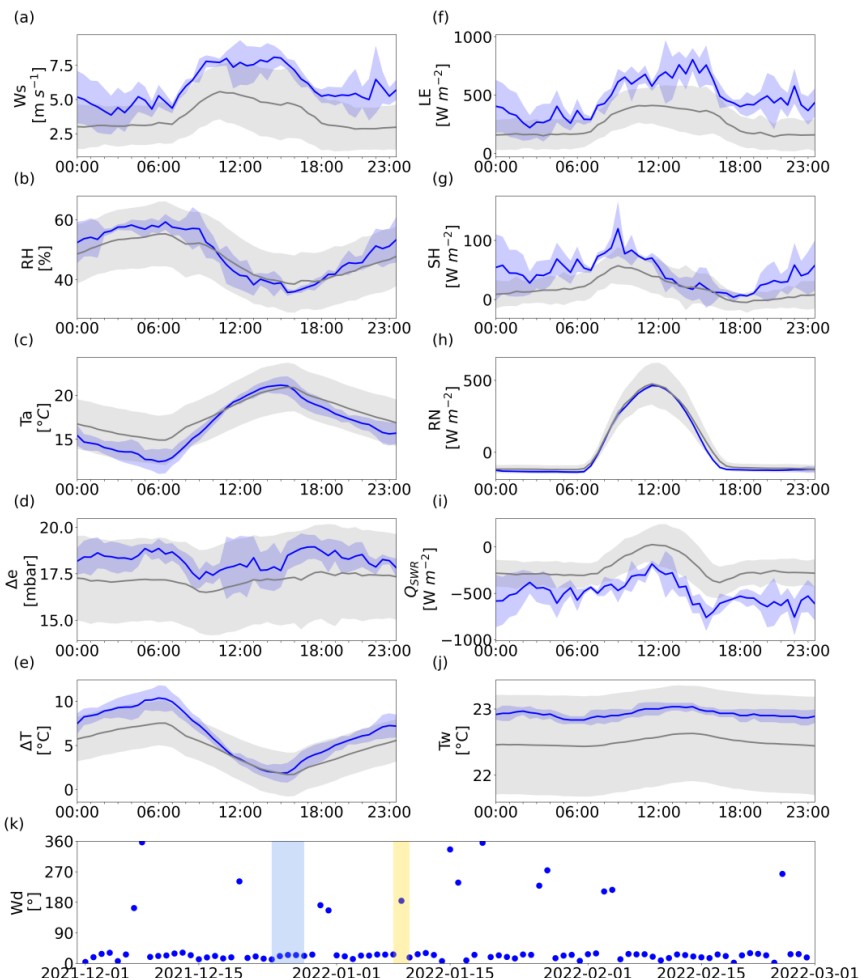

**Figure 8: Extreme surface heat loss case study mean diurnal cycle (LT) during the event (blue) and the reference DJF 2021 period (gray), in the shaded area the +/- 1 SD span. (a) Wind speed ($Ws$), (b) relative humidity ($RH$), (c) air temperature ($Ta$), (d) vapor pressure difference ($\Delta e$), (e) water-air temperature difference ($\Delta T$), (f) latent heat flux ($LE$), (g) sensible heat flux ($SH$), (h) net radiation ($RN$), (i) water temperature ($Tw$), (k) DJF mean daily wind direction, the blue shaded area marks the heat loss event (25-28 December 2021), and the yellow shaded area is the comparison event (9 January 2022) where the wind is sea originated; therefore, high air temperature and relative humidity prevent high rates of heat loss.**

## 4. Discussion

Accurate quantification of heat and gas exchange between semi-enclosed seas and the atmosphere is essential for understanding the exchange of mass, momentum, and energy within the water column. The GoE, a semi-enclosed



sea located in a desert region, is a perfect natural laboratory for studying air-sea interactions in marine environments where stable atmospheric boundary layer persists.

### 4.1 Evaporation rates at the rid semi-enclosed sea: direct measurements vs. bulk formulae

The annual mean evaporation rate of 3.22 m year[-1], presented here, is based on direct measurements and is approximately two times larger than earlier estimations based on indirect bulk formulae (Ben-Sasson et al., 2009). Our findings support the product of $Ws$ and $\Delta e$ as the main forcing mechanism for evaporation in the GoE, in agreement with Lensky et al. (2018) and Tau et al. (2022), with high correlations observed in both diurnal and seasonal time scales. However, in the extremely arid environment of the GoE where $\Delta e$ is always positive and has high values, evaporation is mostly governed by the $Ws$ (correlation of $LE$ to $Ws$ is 0.97, and to $\Delta e \times Ws$ is 0.98). Accordingly, although the mean $\Delta e$ is higher in winter and the water surface temperature is higher than the overlying air temperature, thus promoting thermal convection: Evaporation is higher in summer due to the high wind speed that is sustained longer in the day and the coincidence with the high $\Delta e$ during the afternoon. Ben-Sasson et al. (2009) estimated higher evaporation during winter, based on the bulk formulae approach, which overestimates the role of vertical thermal stability of the air (warmer water underneath the colder air); thus, they predicted that in winter, the free convection due to atmospheric instability led to increased evaporation rates.

The bulk formula estimation of the $LE$ and $SH$ fluxes reproduced similar seasonal trends and values as Ben-Sasson et al. (2009). The comparison to our EC measurements indicated that the bulk formulae are not appropriate for complex desert semi-enclosed seas. Specifically, the parametrization of the ABL stability and the calculation of the friction velocity through their relation to the bulk Richardson number; indicated by the large deviation of the stability parameter and the lower friction velocity of the bulk formula in comparison to the EC values. These findings are consistent with the observations of Bardal et al. (2018) regarding the sensitivity of ABL stability to parametrization methods in coastal regions. Proving to be a major cavity, when stable ABL is calculated using the bulk formula, a lower water vapor transfer coefficient is assigned, resulting in inaccurate lower evaporation rates outside winter. These lower bulk formula evaporation rates occur despite $\Delta e \times Ws$ being higher outside winter, causing also the measured EC evaporation flux to increase outside winter along with $\Delta e \times Ws$.

The complex nature of the sharp gradients between the arid landscape and sea and the proximity to the mountainous shore presents a challenge for the bulk formulae; it was originally calibrated for the open tropical ocean where unstable MABL conditions persist and rely on empirical relations. To address this issue, a similar effort to the TOGA-COARE campaign (Fairall et al., 1996) is required to refine the bulk formulae parameterization in this intricate environment. The efforts need to focus on the relationship between the easily computed bulk Richardson number and atmospheric stability and friction velocity. Until such efforts are made, we recommend that the direct method of the EC approach be used, as it is the only method with sufficient accuracy for evaporation rate estimation in these settings. These lessons will probably transfer to similar regions such as the Persian Gulf and outside the desert regions where coastal upwelling and land originated winds determine a stable ABL over the sea.

The $SH$ flux is small compared to radiation and $LE$ fluxes (-20 W m[-2]) transferring overall heat from the air to the sea according to the air-sea temperature difference. The bulk formula like in the case of $LE$ is underestimating the $SH$ flux.



### 4.2 Diurnal cycle and water temperature regulation

The diurnal cycle of heat fluxes and micrometeorology shows that from mid-spring to mid-autumn in the afternoon the warmer water and the higher air temperature led to reduced $RH$ and consequently to high $\Delta e$. This coincides with the diurnal high $Ws$ which results not only in the daily maximum evaporation rate but also in a double peak mean diurnal evaporation cycle. Furthermore, this results in higher evaporation after maximum diurnal radiation. This timing of the diurnal maximum in the evaporative cooling effect has critical ramifications for the diurnal thermoregulation processes of shallow water, where coral reefs reside. The diurnal energy partitioning demonstrates how an increase in water temperature will be mitigated by evaporative cooling under the normal conditions of sustained daytime winds and extremely dry and hot air. This kind of intense thermoregulation processes can exist in areas where land-sea wind regimes results in persist dry winds flowing over the surface of the sea.

The strong evaporation enabled by the dry air overcoming the input of energy through the surface of the sea, resulting in a negative $Q_{SWR}$ -79 W m$^{-2}$. Similar to the thermoregulation processes at the coral reef of the GoE (Abir et al., 2022) in the summer, the surface fluxes thermoregulate the shallow water temperature at the diurnal time scale. The water temperature reaches its maximum diurnal value shortly after peak radiation, 3 h before the $Ta$ maximum diurnal value. The lag of air temperature after the water temperature is not intuitive, since the heat capacity of water is larger than the air's. This is caused by the strong evaporative cooling effect as indicated by the coincidence of the maximum daily $Tw$ and the crossing of the $Q_{SWR}$ to negative values in the afternoon, emphasizing the effectiveness of $Q_{SWR}$ in analyzing daily surface heat balance and thermoregulation processes.

### 4.3 Energy balance closure and advection term estimation

Throughout most of the 2-year observation period, the surface fluxes cool the surface water. Without the support of advected heat, the evaporation rate will decay due to the cooling of the water and decreased water vapor pressure, therefore this continuous high rate of evaporation can only be sustained with the advection of heat from the south (Biton and Gildor, 2011a).

During the winter, autumn, and summer campaigns, the heat storage was calculated based on temperature measurements of the top 145 m of the GoE. During winter when several hundred meters (>400 m (Shaked and Genin, 2022)) of the water column is mixed and the deficit to complete the energy balance closure could account for mainly two factors; cooling of the deeper water column (which was not measured) and heat transported by advection. During autumn due to the thermocline being deeper than our measurements of temperature (~200 m (Shaked and Genin, 2022)), the large deficit to the energy balance closure (174 W m-2) is attributed to advection through the northern bound surface currents and cooling of the deeper water and deepening of the thermocline. During summer, when the shallow water is strongly stratified and the thermocline is ~100 m deep, and the net surface fluxes are relatively weak (7 W m-2), the warming of the water is a direct result of heat advection from the south as Biton & Gildor (2011b) found. Therefore, we conclude that the surface fluxes contribute to the heating processes of the surface water only at the diurnal scale and offer substantial cooling during the seasonal time scale.

### 4.4 Winter synoptic-scale forcing for high rates of heat loss

We demonstrate how wintertime synoptic scale variability in atmospheric circulation drives extreme rates of heat loss from the water of the GoE. Our direct flux measurements resolved the gap from previous studies, that utilized remote sensing data with larger pixel size than the GoE width (Menezes et al., 2019; Papadopoulos et al., 2013). The steep sea level pressure gradient over the Red Sea, caused by the low pressure system resulted in high wind



speeds which are cold and dry (land originated wind) leading to high rates of heat loss during winter. Cooling of the surface water by the daily mean rate of ~-500 W m$^{-2}$ during the event, 2.5 times larger than the two years winter mean, increased the density of the surface water by increasing salinity through evaporation and by cooling. Thus, the cooling event impacted the vertical stability of the water column and enhances processes that lead to

the annual winter vertical mixing processes. Indeed, the water column at the northern GoE was mixed during that winter (2021/2) to a depth greater than 700 m for the first time in a decade, where the mixing depth didn't exceed ~500 m. Additionally supported by the number of large heat loss events during the winter (especially during December) before the deep mixing year and the shallow depth mixing year in our observation period (>700 and 450 m respectively). This relation of extreme surface heat losses, driven by synoptic-scale circulation, to winter

water column mixing (basin overturning circulation), should be further explored as preconditioning events for winter water column mixing. Along with, how the surface fluxes change with response to winds traversing longer distances over the narrow sea.

**5. Conclusions**

Over a two-year period, direct measurements in the Gulf of Eilat (GoE), locked sea in a hyperarid region, reveal a significantly higher annual mean evaporation rate (3.22 m year$^{-1}$) compared to lower indirect estimates using bulk formulae (1.6-1.8 m year$^{-1}$). Minimum evaporation rate was measured during winter season, due to lowest wind speed and vapor pressure difference. Previous indirect estimations revealed highest evaporation during

winter, due to the highest thermal instability of the ABL.

Higher evaporation rates during the warmer seasons, spring, summer and autumn, are attributed to increased wind speed persisting into the evening, particularly in the afternoon. This aligns temporally with the diurnal maximum in vapor pressure difference, resulting from decreased relative humidity and surface water warming. The seasonal wind speed trend was found stronger than the seasonal vapor pressure difference trend in determining the seasonal

evaporation rate in arid regions.

The current parametrization of bulk formulae vapor transfer coefficient exhibits uncertainties, particularly in stable Atmospheric Boundary Layer (ABL) environments. Our findings underscore the need to revisit ABL stability parametrization and friction velocity calculation, especially in environments where warm air travels over cooler water. The derived vapor transfer coefficient from Eddy Covariance (EC) measurements is 0.00229,

applicable to the northern GoE.

Surface energy exchange fluxes predominantly cool surface water, with minimal heating through the surface fluxes during summer, insufficient to explain annual temperature increase in warm seasons. Advection fluxes, bringing warmer water to the GoE, sustain high evaporation rates year-round. Winter synoptic scale events forcing high surface heat and moisture loss from the sea may precondition deep vertical water column mixing in the GoE.

Accurate evaporation estimates are crucial for ocean circulation modeling both for basic state and acute events modeling, requiring updating ocean dynamics simulations based on the new understanding of air-sea interactions in these environments.

We emphasize the need for future efforts in resolving the spatial distribution of the evaporation rate, ABL stability, and surface friction velocity at the GoE, the Red Sea, and other coastal areas.




## 6. Appendix A

| season | Winter | | | | Spring | | | | Summer | | | | Autumn | | | | Annual | | | |
|---|---|---|---|---|---|---|---|---|---|---|---|---|---|---|---|---|---|---|---|---|
| | mean | std | min | max | mean | std | min | max | mean | std | Min | max | mean | std | min | max | mean | std | min | max |
| $LE$ [W m$^{-2}$] | 232 | 161 | -44 | 1014 | 271 | 173 | -46 | 1070 | 276 | 158 | -49 | 1093 | 241 | 157 | -50 | 994 | 255 | 163 | -50 | 1093 |
| $SH$ [W m$^{-2}$] | 19 | 29 | -239 | 205 | -25 | 47 | -247 | 246 | -49 | 42 | -247 | 244 | -22 | 37 | -249 | 227 | -20 | 46 | -249 | 246 |
| $Ta$[°C] | 18.75 | 3.32 | 9.07 | 28.05 | 24.71 | 5.74 | 10.48 | 42.24 | 32.06 | 3.4 | 22.29 | 43.22 | 27.78 | 4.5 | 16.41 | 42.01 | 25.85 | 6.52 | 9.07 | 43.22 |
| $RH$ [%] | 47 | 12 | 15 | 88 | 37 | 13 | 6 | 82 | 39 | 12 | 9 | 84 | 47 | 12 | 11 | 92 | 42 | 13 | 6 | 92 |
| $Tir$ [°C] | 20.21 | 2.14 | 13.37 | 26.63 | 23.09 | 3.51 | 13.73 | 35.24 | 29.28 | 2.42 | 14.16 | 36.74 | 25.92 | 2.15 | 19.81 | 33.38 | 24.64 | 4.27 | 13.37 | 36.74 |
| $Tw$ [°C] | 22.99 | 0.96 | 21.1 | 25.37 | 22.45 | 0.95 | 19.07 | 26.1 | 26.33 | 1.48 | 23.7 | 30.9 | 26.11 | 1.35 | 22.1 | 30.8 | 24.47 | 2.14 | 19.07 | 30.9 |
| $u$[m s$^{-1}$] | 2.95 | 2.81 | -11.69 | 10.45 | 4.34 | 2.76 | -7.71 | 12.36 | 3.62 | 2.52 | -5.56 | 9.7 | 3.81 | 2.3 | -7.27 | 9.48 | 3.69 | 2.65 | -11.69 | 12.36 |
| $v$[m s$^{-1}$] | 1.33 | 1.93 | -8.12 | 7.03 | 1.24 | 1.75 | -6.51 | 6.92 | 1.66 | 2.74 | -4.74 | 9.34 | 2.15 | 2.3 | -4.42 | 7.89 | 1.59 | 2.25 | -8.12 | 9.34 |
| $DLW$ [W m$^{-2}$] | 323 | 29 | 259 | 477 | 346 | 34 | 271 | 506 | 412 | 40 | 324 | 543 | 381 | 28 | 306 | 473 | 366 | 47 | 259 | 543 |
| $ULW$ [W m$^{-2}$] | 414 | 12 | 376 | 451 | 430 | 20 | 378 | 505 | 465 | 13 | 381 | 508 | 447 | 13 | 411 | 493 | 439 | 24 | 376 | 508 |
| $DSW$ [W m$^{-2}$] | 165 | 242 | 0 | 890 | 279 | 349 | -8 | 1078 | 318 | 376 | -8 | 1042 | 212 | 287 | -3 | 904 | 244 | 324 | -8 | 1078 |
| $USW$ [W m$^{-2}$] | 19 | 20 | -5 | 113 | 17 | 17 | -5 | 119 | 12 | 14 | -5 | 98 | 17 | 17 | -3 | 66 | 16 | 17 | -5 | 119 |
| $RN$[W m$^{-2}$] | 55 | 226 | -147 | 779 | 179 | 337 | -131 | 1031 | 252 | 369 | -118 | 1020 | 129 | 277 | -126 | 838 | 155 | 316 | -147 | 1031 |
| $\Delta e$ [mbar] | 17.31 | 2.63 | 5.34 | 24.92 | 15.58 | 3.01 | 1.83 | 27.19 | 15.77 | 3.94 | 4.26 | 31.89 | 15.86 | 3.51 | 2.06 | 27.95 | 16.13 | 3.38 | 1.83 | 31.89 |
| $\Delta T$ [°C] | 1.46 | 1.65 | -4.47 | 7.07 | -1.62 | 2.39 | -9.62 | 4.47 | -2.78 | 1.5 | -13.04 | 1.5 | -1.86 | 2.63 | -13.04 | 5.48 | -1.21 | 2.63 | -13.04 | 7.07 |
| $\Delta e \times Ws$ [mbar m s$^{-1}$] | 69.73 | 36.42 | 2.37 | 220.67 | 76.48 | 39.03 | 3.12 | 226.88 | 76.55 | 39.26 | 2.59 | 244.64 | 76.08 | 38.1 | 2.9 | 243.3 | 74.72 | 38.33 | 2.37 | 244.64 |
| $\Delta T \times Ws$ [°C m s$^{-1}$] | 5.41 | 7.6 | -27.84 | 44.17 | -8.21 | 13.02 | -65.6 | 22.85 | -14.08 | 10.86 | -93.46 | 7.4 | -10.45 | 15.89 | -99.26 | 26.42 | -6.88 | 14.26 | -99.26 | 44.17 |
| $ET$ [mm day$^{-1}$] | 8.01 | 5.56 | -1.54 | 35.03 | 9.35 | 5.97 | -1.58 | 36.96 | 9.55 | 5.47 | -1.68 | 37.77 | 8.34 | 5.43 | -1.72 | 34.36 | 8.82 | 5.65 | -1.72 | 37.77 |
| $Q_{SWR}$ [W m$^{-2}$] | -195 | 198 | -1033 | 556 | -66 | 297 | -1047 | 946 | 25 | 334 | -978 | 919 | -90 | 239 | -987 | 805 | -81 | 284 | -1047 | 946 |
| $Ws$ [m s$^{-1}$] | 4.11 | 2.21 | 0.3 | 13.34 | 4.98 | 2.51 | 0.35 | 12.98 | 4.91 | 2.32 | 0.2 | 11.93 | 4.89 | 2.44 | 0.35 | 11.4 | 4.72 | 2.4 | 0.2 | 13.34 |





**Table A1: Seasonal mean and the entire observation period statistics of micrometeorological and fluxes variables.**

## 7. Competing interest

The contact author has declared that none of the authors has any competing interests.

## 8. Acknowledgments

We thank the team from the Geological Survey of Israel; Uri Malik, Guy Tau, and Ziv Mor. Yoav Balaban for the graphic illustration. Asaph Rivlin, Modi Pilersdorf, the Israel National Monitoring Program at the Gulf of Eilat, and The Interuniversity Institute for Marine Sciences in Eilat for access to infrastructure and services. Dr. Assaf Zvuloni and Chen Toufikian of Israel's Nature and Parks Authority for their assistance. We thank Prof. Pinhas Alpert and Prof. Hadas Saaroni Lab for giving us access to their daily synoptic classification dataset. We thank the University of Queensland for the financial and technical support of the research. The research was supported by the Israel Science Foundation (Grant ISF-2018/1471), PI–NGL, and by Funds for PIs EM, HM, and NGL from the Hebrew University of Jerusalem–Zelman Cowen Academic Initiatives (ZCAI) Joint Projects 2021 (2022-2024). HG was supported by a grant from the Ministry of Science and Technology.

## 9. Data availability statement

Datasets for this research are available in these in-text data citation references: Abir, S. (2023)

V1. https://doi.org/ 10.17632/wmtdmjgsfp.1.

The IUI monitoring program data used for this research

are available at http://www.meteo-tech.co.il/eilat-yam/eilat_en.asp. The WireWalker CTD data is available upon

request from Prof. Hezi Gildor: Hezi.gildor@mail.huj.ac.il

The ERA5 hourly data on a

single level (Hersbach et al., 2023b) and pressure level (Hersbach et al., 2023a) in this study are available at

https://cds.climate.copernicus.eu/cdsapp#!/dataset/reanalysis-era5-single-levels?tab=overview.

Software for this research is available in these in-text data citation references: LI-COR

Biosciences. (2021). Eddy Covariance Processing Software (Version 7.0.8) [Software]. Available at

www.licor.com/EddyPro.

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
