# Peer review of "Air-sea interactions in stable atmospheric conditions: Lessons"

_EGUsphere, 2023_

## Author Response (AR1)

12 March 2024

Dear Editor, Atmospheric Chemistry and Physics

We thank the editor and two anonymous reviewers for their comments and feedbacks on our revised MS (https://egusphere.copernicus.org/preprints/2024/egusphere-2023-1724/). Below we copied the reviews, in bold, and provide our replies, italicized. We hope that the MS is now ready for publication in the journal of Atmospheric Chemistry and Physics.

Sincerely,

Shai Abir (first author) and Nadav Lensky (corresponding author), on behalf of the co-authors.

Reply to Anonymous referee #1 (RC2):

**The authors addressed my comments, provided satisfactory answers, and incorporated these changes in the updated manuscript . It is good to go from my side..** (https://doi.org/10.5194/egusphere-2023-1724-RC2).

*Thank you for your meticulous review and invaluable feedback on our manuscript, your insights have greatly enhanced the quality of our work.* (https://doi.org/10.5194/egusphere-2023-1724-AC1).

Reply to Anonymous referee #2 (RC1):

*We are grateful to the referee for the insightful comments and valuable suggestions ( https://doi.org/10.5194/egusphere-2023-1724-RC1), which have guided our response and clarified several aspects of our manuscript (our replies: https://doi.org/10.5194/egusphere-2023-1724-CC1).*

**The results fully correspond to what one would expect for the region, so that no new methodological or process-based findings emerge.**

*We disagree with the reviewer on the statement that the results are fully expected for the region; actually, as we demonstrate, our results contradict previous estimates of air-sea fluxes. As far as we know, there has never been direct long-term measurements by Eddy covariance in the studied region in particular or in any desert semi-enclosed seas to quantify evaporation and its impacts on marine thermo-regulation. Previous studies of air-sea heat exchange, including latent heat fluxes over desert seas relied on bulk formulae, which our manuscript shows deviate fundamentally from the direct observations of heat exchanges. The bulk formulae, which are widely utilized, are important, as it is often not feasible to make direct measurements of air-sea energy exchanges, while these processes must be represented in numerical models by bulk*

*formula. Accordingly, there is great value in improving them, which can only be done by validating their indirect estimation with direct measurements like ours. In this manuscript, we present a unique set of direct measurements, using a flux tower over the Gulf of Eilat (eddy covariance, radiometers, thermistors), continuously measuring over two years, and we present the actual diurnal, seasonal, and annual cycles of evaporation and heat flux partitioning. We analyze our flux measurements and their relations to the changes in the environment (diurnal and seasonal cycles). Our results, e.g., diurnal, seasonal, and annual evaporation flux, deviate significantly from "what one would expect for the region"; in Section 2.1 we present the scientific literature in this topic and the large uncertainties in previous studies that were based on estimating evaporation using bulk formulae. In this study, we show that the annual evaporation rate is ~**3.2 m**, i.e. about twice the previous estimations (Section 3.2.2). Furthermore, previous studies estimated the highest evaporation rates during winter, due to an unstable atmospheric boundary layer, but we show that the highest rates are in summer. In Section 3.2.4 we present the deviations of bulk formulae from the directly measured evaporation with emphasis on the large deviation during winter, and in Section 4.1 we discuss the physical mechanisms that force these deviations, such as overcoming of the local micrometeorology over the stability of the boundary layer. We have emphasized these non-trivial aspects in the 'plain language summary', the 'abstract', and throughout the manuscript.*

It would have been possible to discuss some of the processes reflected in the data in more detail, e.g.

- Advection of dry desert air and formation of the clearly visible oasis effect in summer, or the unstable stratification in winter,

*As the reviewer mentioned there is a clear oasis effect during summer which in our paper is characterized by the cooling of the surface water by strong evaporation rates caused by dry, hot air blowing above the cooler sea surface. The oasis effect and its importance for the marine biological system of the Gulf of Eilat have been described in length in our previous paper Abir et al. (2022). Following the reviewer's comment, we now include the term 'oasis effect' in the introduction, where we refer to the paper of Abir et al. (2022) and mention it in the discussion (Sections 4.2). In this manuscript, we present the same oasis effect for longer periods and with additional insights regarding the Gulf's circulation (Section 4.3), we added this terminology in the discussion as well.*

*The unstable stratification of the atmospheric boundary layer (referred to in the manuscript as unstable atmospheric boundary layer) in winter was previously recognized as a major driver of higher evaporation rates in winter, according to the bulk formulae, as mentioned above and as presented in the introduction (e.g., Ben Sasson et al. 2009). In this manuscript, we show that the wind speed in winter is lower than in the warmer seasons and conclude that in extremely dry areas wind speed seasonal trend overcomes the effect of thermal stratification instability, which promotes convection in the atmospheric boundary layer. For more details, see Sections 3.2.4, Section 4.1, and Figure 5.*

- Influence of the proportion of land in the footprint with northerly winds

*As can be viewed in Figure 1d, there was minimal contribution of land-originated winds due to the dominant winds that are from the sea surface, and minor land-derived winds were filtered out, as described in Sections 2.2.2 and 2.2.3.*

- Possible local circulations that reinforce the large-scale (synoptic) circulation.

*Local circulation that reinforces the large-scale circulation is a very interesting topic and can have an important role in air-sea heat and gas exchange fluxes. At the Gulf of Eilat some of these reinforcement mechanisms such as the sea breeze where previously described by Saaroni et al. (2004). However, it is beyond the scope of this research.*

- Fig 1d is probably meant to represent a footprint climatology, however, it is incomprehensible when compared to the original literature or textbooks (Amiro, 1998;Leclerc and Foken, 2014). In addition, EddyPro offers several footprint models, the one used should already be named.

*We thank the reviewer for noting this issue. We clarified that in the revised text in Section 2.2.2. Eddypro uses three models to calculate the footprint:*

- *Kljun et al. (2004): A crosswind integrated parameterization of footprint estimations obtained with a 3D Lagrangian model by means of a scaling procedure.*

- *Kormann and Meixner (2001): A crosswind integrated model based on the solution of the two-dimensional advection-diffusion equation given by van Ulden (1978) and others for power-law profiles in wind velocity and eddy diffusivity.*

- *Hsieh et al. (2000): A crosswind integrated model based on the former model of Gash (1986) and on simulations with a Lagrangian stochastic model.*

*In the output file and the dataset for this manuscript (Abir et al. 2023) there is a column (column header='footprint') specifying the footprint model chosen by EddyPro, coded (0) for Kljun et al. (2004), (1) Kormann and Meixner (2001) and (2) for Hsieh et al. (2000). The Kljun model was used almost exclusively by the EddyPro for our research.*

- To determine the data quality, reference is made to the paper by Foken et al. (2004). However, the flag=2, which was excluded from further processing by the authors, indicates particularly good data. EddyPro offers summarised flags. The authors have obviously used a flag system with only 2-3 levels, but then this must also be cited.

*We thank the reviewer for this remark on the citation. The Foken flags are indeed processed by EddyPro to 3 levels system by Mauder and Foken (2004). We clarified that in the revised manuscript, in section 2.2.2.*

- The separation of sonic anemometer and gas analyser of almost 0.5 m is considerable. Here it would be important to know which method offered by EddyPro was used to correct the spectral losses in the high-frequency range.

*We clarified that in Section 2.2.2 and Section 2.2.3. The EddyPro default setting for express mode and the associated spectral correction (Moncrieff et al. 1997, 2004) which also correct for the separation of the IRGA and wind anemometer can be found here (citied in the manuscript) under spectral correction: https://www.licor.com/env/support/EddyPro/topics/express-defaults.html*

- It is not entirely clear whether a bulk method was also used to determine the Obukhov length and the friction velocity (Fairall et al., 1996), although the data are also available using the eddy covariance method.

*We clarified the two different ways of calculations in Section 2.2.5. The Obukhov length and friction velocity were obtained by the two methods (see SI section on calculation atmospheric stability): (1) by the eddy covariance method utilizing the 3D wind anemometer, and (2) from the calculations of the bulk model (COARE3.6) (Fairall et al., 1996). This is a key comparison we perform in the manuscript (Figure 5) in order to point the origin of the discrepancy of the bulk model evaporation estimates from the directly measured eddy covariance measurements. See more in Section 3.2.4 and Section 4.1.*

*References:*

*Abir, S., McGowan, H., Shaked, Y., & Lensky, N. (2022). Eddy covariance measurements over the Gulf of Eilat (Aqaba) fringing coral reef, V3 [Dataset]. Mendeley Data. https://doi.org/10.17632/wsfrgbd5t8.3*

*Abir, S., McGowan, H., Shaked, Y., Morin, E., and Lensky, N. (2023). 2 years Eddy Covariance measurements over the Gulf of Eilat (Aqaba), V1 [Dataset], https://doi.org/10.17632/wmtdmjgsfp.1.*

*Ben-Sasson, M., Brenner, S., and Paldor, N.: Estimating air-sea heat fluxes in semienclosed basins: The case of the Gulf of Elat (Aqaba), J Phys Oceanogr, 39, 185–202, https://doi.org/10.1175/2008JPO3858.1, 2009.*

*Fairall, C. W., Bradley, E. F., Rogers, D. P., Edson, J. B., and Young, G. S.: Bulk parameterization of air-sea fluxes for tropical ocean-global atmosphere coupled-ocean atmosphere response experiment, Journal Geophysical Research Oceans, 101, 3747–3764, https://doi.org/10.1029/95JC03205, 1996.*

Mauder, M., Foken, T: Documentation and instruction manual of the eddy covariance software package TK2. Universität Bayreuth, Abt. Mikrometeorologie, Arbeitsergebnisse 26, 44 pp. (Print: ISSN 1614-8916; Internet: ISSN 1614-8926), 2004.

Moncrieff, J. B., Massheder, J. M., de Bruin, H., Ebers, J., Friborg, T., Heusinkveld, B., Kabat, P., Scott, S., Soegaard, H., and Verhoef, A.: A system to measure surface fluxes of momentum, sensible heat, water vapor and carbon dioxide, Journal of Hydrology., 188-189, 589–611, https://doi.org/10.1016/S0022-1694(96)03194-0, 1997.

Moncrieff, J. B., Clement, R., Finnigan, J., & Meyers, T.: Averaging, detrending and filtering of eddy covariance time series, in Handbook of Micrometeorology: A Guide for Surface Flux Measurements, edited by Lee, X., Massman, W. J., & Law, B. E., Kluwer Academic, Dordrecht, 7–31, 2004.

Saaroni, H., Maza, E., & Ziv, B.: Summer sea breeze, under suppressive synoptic forcing, in a hyper-arid city: Eilat, Israel. Climate Research, 26(3), 213–220. https://doi.org/10.3354/cr026213, 2004.